# *Brucella* Phagocytosis Mediated by Pathogen-Host Interactions and Their Intracellular Survival

**DOI:** 10.3390/microorganisms10102003

**Published:** 2022-10-11

**Authors:** Tran X. N. Huy, Trang T. Nguyen, Heejin Kim, Alisha W. B. Reyes, Suk Kim

**Affiliations:** 1Institute of Animal Medicine, College of Veterinary Medicine, Gyeongsang National University, Jinju 52828, Korea; 2Institute of Applied Sciences, HUTECH University, 475A Dien Bien Phu St., Ward 25, Binh Thanh District, Ho Chi Minh City 72300, Vietnam; 3Department of Veterinary Paraclinical Sciences, College of Veterinary Medicine, University of the Philippines Los Baños, Laguna 4031, Philippines

**Keywords:** *Brucella*, phagocytosis, adhesin, receptor, intracellular trafficking, phagolysosome fusion

## Abstract

The *Brucella* species is the causative agent of brucellosis in humans and animals. So far, brucellosis has caused considerable economic losses and serious public health threats. Furthermore, *Brucella* is classified as a category B bioterrorism agent. Although the mortality of brucellosis is low, the pathogens are persistent in mammalian hosts and result in chronic infection. *Brucella* is a facultative intracellular bacterium; hence, it has to invade different professional and non-professional phagocytes through the host phagocytosis mechanism to establish its lifecycle. The phagocytosis of *Brucella* into the host cells undergoes several phases including *Brucella* detection, formation of *Brucella*-containing vacuoles, and *Brucella* survival via intracellular growth or being killed by host-specific bactericidal activities. Different host surface receptors contribute effectively to recognize *Brucella* including non-opsonic receptors (toll-like receptors and scavenger receptor A) or opsonic receptors (Fc receptors and complement system receptors). *Brucella* lacks classical virulence factors such as exotoxin, spores, cytolysins, exoenzymes, virulence plasmid, and capsules. However, once internalized, *Brucella* expresses various virulence factors to avoid phagolysosome fusion, bypass harsh environments, and establish a replicative niche. This review provides general and updated information regarding *Brucella* phagocytosis mediated by pathogen-host interactions and their intracellular survival in host cells.

## 1. Introduction

Brucellosis is a globally distributed major bacterial zoonosis characterized by abortion and reproductive failure in livestock, which seriously affects the development of animal husbandry and international trade. Although the mortality rate is low, it is harmful to human health as it can lead to a debilitating disease and serious chronic complications if left untreated [1,2,3]. The transmission of infection to humans is primarily via direct contact with animals, handling of contaminated tissues, and consumption of unpasteurized milk [1]. Furthermore, the *Brucella* species is classified as a category B bioterrorism agent due to its ease of transmission via aerosol [4].

No licensed human vaccine is available against brucellosis, and despite harsh therapy using a combination of two antibiotics for several weeks to months, relapses or clinical failures still occur [5,6]. Consequently, the control of human brucellosis depends on the control of brucellosis in livestock, which cannot primarily rely on vaccination alone but must be tackled in combination with proper husbandry measures [6,7]. Additionally, the widely used live attenuated *Brucella* vaccines for animals, such as *Brucella* (*B.*) *abortus* S19 and *B. melitensis* Rev. 1, both induce abortions, are virulent for humans, and interfere with serodiagnosis. Furthermore, Rev. 1 is resistant to streptomycin, which is a necessary antibiotic used for the treatment of the disease, and *B. abortus* RB51 still presents residual virulence [6,8].

The causative agent, *Brucella* species, is a Gram-negative facultative intracellular bacterium. Among *Brucella* species, three of them are known to be endemic in most countries and are highly virulent to both their natural hosts and to humans, including *B. abortus, B. melitensis* and *B. suis* that primarily infect cattle, sheep and goats, and domestic feral and wild swine populations, respectively [4,9]. *Brucella* lacks well-known or classical virulence factors such as spores, fimbriae, cytolysins, exotoxins, secreted proteases, antigenic variation, resistance forms, phage-encoded toxins, virulence plasmids, and capsules [7,10]. The virulence factors that have been reported to be necessary for invasion, the establishment of infection, as well as intracellular survival and replication of *Brucella* are cyclic β-1,2-glucan (CβG), VirB T4SS, pathogen-associated molecular patterns, two-component sensory and regulatory system BvrS/BvrR, and lipopolysaccharide (LPS) [2,7]. *Brucella* LPS is altered, so it is a weak inducer of the host inflammatory response compared to LPS molecules of other Gram-negative bacteria [9,11]. Other virulence factors include outer membrane proteins, BacA, SagA, BmaC, BetB, BtaE, MucR, and a genomic island associated with *Brucella* pathogenicity [7].

*Brucella* can infect both professional phagocytes such as macrophages and dendritic cells (DCs), and non-professional phagocytes such as placental trophoblasts and epithelial cells [6,11]. Other cells can also be infected by *Brucella,* such as neutrophils, lymphocytes, and erythrocytes, although no efficient intracellular replication is attributed as they are more associated with bacterial dispersion hence providing an indication of their regulatory role in the persistence of the bacteria [1]. Due to its capacity to survive and replicate within host macrophages, the *Brucella* pathogen has the ability to produce chronic infection that could lead to life-long infections. Dendritic cells (DCs) are well-known antigen presenting cells, which are also considered a safe haven for *Brucella* growth. *Brucella* can interfere with their maturation leading to the inhibition of antigen processing and presentation that circumvents the host immune response. Moreover, *Brucella* can infect the animal placenta resulting in abortion. In particular, it can replicate in the placental trophoblasts, where erythritol is produced. Indeed, the erythritol utilization is one of *Brucella*’s virulence factors [11]. Four steps are essential for *Brucella* to infect the host: adherence, internalization, intracellular growth, and dissemination within the host [7]. This review mainly focuses on the phagocytosis of *Brucella* spp. into the host cells as well as its intracellular growth in the macrophage cell model, on which there is currently still minimal information available to completely describe *Brucella*’s interaction with its target cells and tissues.

## 2. *Brucella* Phagocytosis and Intracellular Survival

### 2.1. Pathogen Recognition

#### 2.1.1. Bacterial Adhesion

As mentioned earlier, *Brucella* possesses the ability to cause chronic infection in many different cell types. *Brucella* must break the epithelial barriers to cause systemic infection. To do that, the bacteria first have to invade the host cells to initiate their intracellular infectious cycle. Therefore, adhesion to the host cells is the first and most essential step in the invasion process. *Brucella* expresses various adhesin molecules to mediate the host and pathogen interaction. *Brucella* adhesins include the sialic acid-binding proteins SP29 and SP41; BigA and BigB proteins containing the immunoglobulin-like domain; the monomeric autotransporters BmaA, BmaB, and BmaC; the trimeric autotransporters BtaE and BtaF; and collagen, vitronectin-binding protein Bp26. These molecules regulate *Brucella* adhesion to host cell surface molecules and extracellular matrix components. In addition to ensuring the adhesion of *Brucella* to the host cell surface, these adhesins are considered immunogens or vaccine candidates to activate host immunity [12,13,14,15,16,17]. A study by Castaneda-Roldan et al. [16] displayed that a *Brucella* surface protein SP41 can bind selectively to epithelial HeLa cells. This adhesin protein is the expressed product of the *ugpB* gene. *Brucella ugpB* knockout strain decreased bacterial internalization by 40- to 50-fold less than the wild-type strain. Besides, another surface protein with an apparent molecular mass of 29-kDa was demonstrated to target the host gangliosides structure on red blood cells. A fraction of SP29 was found as a periplasmic protein in *B. melitensis* [18]. A region in chromosome 1 of *B. abortus* contains a gene, *bigA*, which codes an adhesin. This adhesin mediates the adherence and invasion into Madin-Darby canine kidney (MDCK) and Caco-2 cells. By performing immunofluorescence microscopy and Western blot assay, this adhesin was proven to localize at the outer membrane and contain an immunoglobulin-like domain, respectively [19]. It is noteworthy that VirB5 is an effector of a well-known *Brucella* virulence factor T4SS that plays an essential role in *Brucella* infection and is also recognized as an adhesin to contact with the host cells. Indeed, Deng et al. [20] determined that a single domain antibody, BaV5VH4, can bind *Brucella* VirB5 protein, disrupting the interaction between *Brucella* and host macrophage cells. Though the VirB5 protein in *Agrobacterium tumefaciens* localizes to the T-pilus tip that contributes to host-cell recognition, its direct adhesion function in this bacterium or *Brucella* is still unclear and needs to be deeply clarified [21].

#### 2.1.2. Host Receptors

Once *Brucella* adheres to host cells, the binding of the pathogen to various host cell receptors activates a series of signaling pathways essential for bacterial uptake. Opsonized bacteria are internalized via complement receptors (CRs) and Fc receptors (FcRs) which promote the efficient phagocytosis of *Brucella* into host phagocytes. A major virulence factor of *Brucella*, the LPS O-chain fragment, is responsible for binding to antibodies or the C3 component of complement systems [22,23,24]. Opsonization of *Brucella* enhances the uptake of *Brucella* into human monocytes [25]. Eze et al. [22] reported that antibody and complement-mediated opsonization resulted in a profound increase in the phagocytosis of *B. melitensis* in murine peritoneal macrophages in individual or synergistic effects. However, another study by Rittig et al. [26] showed that the opsonization of *Brucella* by antibody but not complement contributed to increasing the uptake of *Brucella* in human monocytes. These authors proposed a reason that complement receptor 3 (CR3) has both opsonic and non-opsonic binding sites. Thus, *Brucella* can bind directly to this receptor without the need for the main opsonic complement fragment iC3b [26,27,28]. Although the opsonic entry plays an essential role in the early stage of *Brucella* infection, studies have reported that the contribution of FcRs and CRs in the phagocytosis of *Brucella* are still limited. Among FcRs, FCγRIIA plays a crucial role in the phagocytosis of IgG2-opsonized bacteria. Hosseini khah et al. [29] proved that FCγRIIA (CD32) and the polymorphism of this receptor are associated with susceptibility to brucellosis in humans. The most efficient phagocytic receptor among CRs is CR3 (integrin, Mac-1) [28,30].

On the other hand, host cell receptors classified as lectin and fibronectin receptors recognize non-opsonized bacteria. Scavenger receptor A (SR-A) is a well-described lectin receptor in recognition of *Brucella*. This receptor can effectively detect lipid A fraction of LPS [31]. Interestingly, following *Brucella* internalization by SR-A recognition, T4SS is considered a regulator for SR-A signal transduction that enables *Brucella* to establish an intracellular lifecycle in the host cells [31,32].

Among host receptors that recognize *Brucella*, toll-like receptors (TLRs) are most characterized for their functions in recognition and immune response against brucellosis [33]. Much experimental evidence suggests the essential role of TLRs in phagocytosis and in the host signaling pathways to protect against *Brucella* infection. TLR2, TLR4, and TLR6 are supposed to recognize different membrane components of *Brucella*, while TLR3, TLR7, and TLR9 have been found accountable for the detection of *Brucella* nucleic acid structure motifs [34,35,36,37,38]. Although TLRs contribute to the host protection against *Brucella* infection, these pathogens may interfere with TLRs-mediated immune signaling to evade recognition by the innate immune system. A TIR domain-containing protein in *Brucella* (TcpB) interferes with MAL-TLR4 interaction leading to the inhibition of TLR4 signaling pathway. Particularly, the DD loop of TB-8 and TB-9, which are TcpB-derived decoy peptides, is supposed to be able to specifically bind to the MAL TIR domain to form a dimer, then disrupt MAL TIR domain-mediated interaction leading to the inhibition of TLR4 signaling. This interference prevents DCs maturation and the cytotoxic activity of cytotoxic T lymphocytes during *Brucella* chronic infection [39,40,41]. TLR2 recognizes *Brucella* lipoprotein, but it has been revealed to have no function in controlling *Brucella* infection in the TLR2 knockout mice model [42,43]. TLR4 can recognize LPS, and a non-canonical lipid A of LPS at a very high concentration resulting in the promotion of the maturation phagocytic vacuoles and inducing the pro-inflammatory cytokine TNF-α [44]. Regarding receptor costimulation, *Brucella* can express more than one ligand for various host cell receptors. Heat-killed *B. abortus* simultaneously costimulates TLR2 and TLR9 in mouse DCs. This costimulation ensures bacterial phagocytosis and promotes the downstream signal transduction of these receptors [45]. In addition, TLR2 also plays a critical role in *Brucella* invasion into trophoblast giant cells [46]. This suggests that *Brucella* can infect different types of cells in the same way.

### 2.2. Engulfment and Internalization Process Activation

#### 2.2.1. Bacterial Engulfment

Upon translocation across the mucosal epithelial cell layer, professional phagocytes such as macrophages, DCs, and macrophage-like cells engulf the bacteria in which <10% of these phagocytosed *Brucella* survive and escape killing during the initial phase of infection. Following the binding of *Brucella*, phagocytic and non-phagocytic receptors mediate a variety of signaling pathways, leading to the remodeling of the cell membrane structure to engulf bacteria [6,8]. Generally, *Brucella* induces a zipper-like mechanism for internalization [47]. Up to 8 min after contact, *Brucella* moves in a swimming motion on the cell surface with a generalized membrane ruffling. At the same time, the cell surface is rearranged and actin polymerization is activated around the side of bacterial adherence. Notably, these events depend on the *Brucella* virulence factor VirB [48,49]. Moreover, the cellular prion protein (PrP^c^) is a glycoprotein anchored on the outer leaflet of the plasma membrane. It is associated with phagocytosis via cytoskeletal rearrangement and the host inflammatory response. The PrP^c^ is described as a major internalization receptor on M cells during *Brucella* via the oral route [50,51,52]. However, the role of PrP^c^ in *Brucella* internalization is still controversial. Indeed, upon *B. melitensis* infection, silencing of PrP^c^ in microglia cells did not affect bacterial phagocytosis and intracellular killing [53].

Actin polymerization is required in the penetration of *Brucella* into host cells. Bacterial adhesion to cell surface involves other proteins acting as second messengers including cGMP, PIP3-kinase, MAP-kinase, and tyrosine kinase that leads to activation of GTPases of Rho subfamilies such as Rho, Rac, and Cdc42, which play a critical role in the regulation of the cytoskeleton [8,54]. Additionally, a genome-wide small interfering RNA perturbation screen was performed to identify novel host factors crucial for *Brucella* intracellular trafficking. The results displayed that the prominently affected clusters are related to actin-remodeling and phagocytosis. These clusters include ARP2/3, WASP regulatory complex, and the small GTPases Rac1 and Cdc42 [55]. Consistently, adhesion to the macrophage surface involves the activation of GTPases and F-actin polymerization where, at the early stage of infection, annexin I is also implicated in membrane fusion [8,49,54]. Besides, adhesin protein BigA was proven to induce actin-cytoskeleton rearrangement in MDCK and Caco-2 cell lines [19]. The TLR4/PI3-kinase signaling pathway is essential for smooth *Brucella* strain internalization but not in the rough strain. The smooth strain infection quickly elicits F-actin polymerization at the early stage of infection and fuses the early endosome faster than the rough strain [56].

#### 2.2.2. Formation and Maturation of *Brucella*-Containing Vacuoles

Internalization occurs via lipid raft microdomains at the macrophage cell membrane that are rich in cholesterol, glycosylphosphatidylinositol, and gangliosides, and contribute to the intracellular trafficking of *Brucella* [8,52]. Following interaction with the cell membrane of professional or non-professional phagocytes and around 5 min after entry, BCVs enter the endocytic pathway, initiated by the fusion with early endosome, characterized by Rab5, early endosomal antigen 1, and the transferrin receptor (TfR) markers [57,58]. At the next stage of internalization, BCVs continue to fuse with the late endosomes that undergo transient acidification, leading to changes in bacterial gene expression and intracellular survival associated with the late endosomal markers Rab7, RILP, and Lamp1 during 4 h post-infection (pi) [55,59,60]. The presence or absence of these lysosomal markers in the BCVs maturation can be used to evaluate the effect of *Brucella* virulence factors on its intracellular survival. *Brucella* expresses various effector molecules to modulate its intracellular trafficking. For example, *Brucella* can secrete a lysozyme-like protein SagA identified as a muramidase. This enzyme participates in the early stage of *Brucella* intracellular trafficking by avoiding BCVs and lysosome fusion. In particular, it regulates the recruitment of Lamp1 in the BCVs maturation by an unknown mechanism [61]. In addition, *Brucella* membrane fusogenic protein interacts with phospholipid vesicles to favor membrane fusion and is involved in the recruitment of lysosomal markers, Lamp1 [62]. However, the specific mechanism in this event is still unclear [63]. A study by Naroeni et al. [64] revealed that live *B. suis* prevented the fusion of *B. suis*-containing phagosomes with the lysosome compartment while this association was clearly observed using killed *B. suis*.

The interaction between BCVs and endoplasmic reticulum (ER) from 4 to 8 h pi results in the formation of replicative BCVs (rBCVs) with the presence of several ER markers such as calnexin, calreticulin, and Sec61β [60]. It is noteworthy that interaction between lipid rafts and O-chain of smooth *Brucella* LPS plays a vital role in the mediation of the bacterial entry that leads to the development of specialized membrane-bound compartments known as replicative phagosomes or replicative vacuoles [65]. The continuity of rBCV membranes with the ER was observed unmistakably by applying the 3D-CLEM approach. This study proposed a model where rBCVs are integrated into the ER meshwork [66]. On the other hand, CβG is well-known as a virulence factor of *Brucella* sp. and can modulate lipid raft organization. CβG probably acts on lipid raft to favor the recruitment of lipid raft marker flotillin-1 from 0.5 to 8 h pi. In addition, CβG regulates the lysosome fusion and enables BCVs to associate with ER [67,68].

### 2.3. Host Intracellular Killing or Bacterial Intracellular Growth

#### 2.3.1. Host Bactericidal Effect against *Brucella*

Following bacterial engulfment by host cells, most ingested *Brucella* that are localized in the endosomal BCVs (eBCVs) are killed within the phagolysosome compartment in the last stage of phagosome maturation during 4 to 8 h pi. Using a fluorescent probe specifically for proteolytic activity, during the first 6 h pi, the BCVs clearly showed fluorescent intensity emission, and this remarkable fluorescence was not detected at 12 h pi [69]. The phagolysosome compartment displays many effectors that limit *Brucella* growth and survival such as reactive oxygen species (ROS), nitric oxide (NO), and antimicrobial peptides [70,71]. Nitric oxide and lysozyme activities are employed to monitor RB51 vaccine efficiency in cows [72]. This is because NO can react with structural elements, metabolic enzymes, nucleic acid, and it inhibits bacterial secretion system expression. A study by Hu et al. [73] indicated that thioredoxin-interacting protein (TXNIP), a multifunctional protein in metabolic diseases, is involved in the production of NO and ROS, leading to the control of bacyccterial intracellular survival. These authors proposed a novel host pathway, iNOS/NO-TXNIP-iNOS/NO, related to *Brucella* T4SS expression. Aside from NO, although ROS causes DNA, protein, and lipid damage, *Brucella* possesses *xthA-1* gene that encodes a protein that contributes to oxidative stress resistance [74]. On the other hand, rough *Brucella* induces monocytes to secrete higher CXC, CC chemokines (GRO-α, IL-8, MCP-1α, MCP-1β, MCP-1, and RANTES), pro-inflammatory (IL-6, TNF-α), and anti-inflammatory (IL-10) cytokines than the smooth strain [75]. The lacking of the O-chain of LPS interferes with the fusion between the BCVs and lysosomes [76]. Therefore, the decrease in bacterial viability is more dramatic with strains that are lacking in the O-chain of LPS.

#### 2.3.2. *Brucella* Intracellular Survival

In the early stage of phagocytosis, a small proportion of BCVs allows *Brucella* to start proliferating. Some evidence for this event include the inhibition of Rab7 recruitment, which is essential for phagolysosome fusion, and the acidification of this compartment, which triggers the expression of T4SS. Besides, *Brucella* chromosomal replication was observed at 8 h pi within Lamp1-positive BCVs [77]. Up-to-date studies revealed that effectors of T4SS have crucial roles in inhibiting host immune responses and promoting intracellular trafficking and growth during *Brucella* infection. In particular, RicA can interact with Rab2 GTPase, which is critical for *Brucella* intracellular trafficking [60]. VceC and VecA are related to host autophagy and apoptosis. BtpA and BtpB are required to modulate host immunity and energy metabolism. In addition, other T4SS effectors also play different roles in the host-*Brucella* interaction that have been extensively studied, clarifying their specific functions [78,79,80]. A study by Martinez-Nunez et al. [81] demonstrated that BvrS/R transcriptionally regulates T4SS VirB. A *bvrS* and *bvrR* mutant resulted in low levels of the VirB1, VirB5, VirB8, and VirB9. In addition, this study clarified that the regulator protein BvrR binds directly to the VirB promoter. The BvrR/BvrS system is one of the virulence factors that interacts with other diverse virulence factors simultaneously to ensure the intracellular growth fate of *Brucella*.

*Brucella* can start multiplying in macrophages at 12 to 24 h pi [69]. At this time, two events are in progress. First, the pathogen readily reaches its safe haven to replicate. Lastly, the pathogen has set up its specific virulence genes. The conversion from eBCVs to rBCVs is a hallmark of *Brucella* intracellular growth. In addition to the beneficial functions of T4SS in promoting *Brucella* intracellular replication, a study by Casanova et al. [55] discovered two components of the trimeric vacuolar protein sorting complex, VPS35 and VPS26A, which are related to the evasion of the lysosomal degradative pathway. In this infection stage, *Brucella* is in a multi-membranous compartment, supplemented with several autophagosomal bodies and ER markers. The association of *Brucella* with the host ER establishes the replicative niche. By reaching this organelle, *Brucella* can take advantage of the biosynthetic enzymes, connect to the local nutrient supply, and avoid phagolysosome fusion [76,77,82]. On the other hand, Kohler et al. [76] proposed the term “brucellosome” for the replicative niche of *Brucella,* which is poor in nutrients, low in oxygen tension, and with a neutral pH microenvironment. However, *Brucella* still furtively exploits these harsh environments as stimuli to induce virulence genes involved in their intracellular trafficking and growth [83]. Once adapted to the intramacrophage environment, *Brucella* extends its intracellular persistence indefinitely, contributing to systemic proliferation and dissemination to other targeted cells or tissues. Furthermore, at this late stage of infection, the switch from rBCVs to the autophagic BCVs (aBCVs) mediates bacterial release from infected cells. Autophagy-initiation proteins ULK1, Beclin 1, ATG14L and PI3-kinase activity is required for the aBCVs formation. This event is exploited by *Brucella* for cell-to-cell spreading [84,85,86]. However, once again, *Brucella* continues displaying an effector of the T4SS system, BspL, to delay the formation of aBCVs that benefit the optimal intracellular replication before disseminating to other cells. At the same time, BspJ, a nucleomodulin, directly or indirectly regulates host cell apoptosis to complete its intracellular cycle [87,88]. As mentioned, the microenvironment inside these vacuoles has limited nutrients. Interestingly, *Brucella* behaves as a stealthy pathogen. It has adapted to these harsh intracellular environments in the host cells by setting up various virulence factors such as the heat shock proteins DnaK and ClpB. These virulent factors play an important role in *Brucella* resistance against some stresses but do not favor intracellular growth [89].

## 3. Conclusions

*Brucella* sp. is a facultative intracellular pathogen. It can infect professional and non-professional phagocytes. To establish its intracellular niche, it has to invade host cells, initiated by bacterial adherence to the host cell surface. The detection of *Brucella* by various host cell receptors is the second phase of bacterial phagocytosis (Figure 1). Once *Brucella* is detected by either opsonic or non-opsonic receptors, various signaling pathways are activated to engulf and uptake it into the host cells. In the typical way of phagocytosis, BCVs enter the endosomal pathway from the fusion steps with early endosomes, late endosomes, and lysosomes resulting in bacterial elimination. Nevertheless, as a stealthy pathogen, *Brucella* facilitates its intracellular replication by exploiting host cell resources. More than that, *Brucella* has various effectors that interfere with antibacterial mechanisms to ensure its intracellular survival (Figure 2). Furthermore, all *Brucella* functional proteins related to the interaction between *Brucella* and host cells is summarized in Table 1.

Brucellosis is a worldwide disease causing severe economic and public health problems. Therefore, understanding the characteristics of the causative agent, its interaction with host cells and specific virulence factors is vital in eradicating brucellosis. Several recent studies focus on understanding the interaction between *Brucella* and its host, and the mechanism of how it works, but still the available data are limited and the complete mechanisms involved are unclear. In fact, in comparison to other bacteria such as *Mycobacterium* species, *Salmonella* species, and *Helicobacterium* species, more research has been recently invested in *Brucella* species. However, there are still many gaps in understanding *Brucella* phagocytosis, and many questions remain unanswered particularly regarding the interaction between molecules related to F-actin polymerization including cofilin, profilin, ARP2/3, and WASP that renders importance in the context of *Brucella* infection. In addition, searching for more virulence factors implicated during *Brucella* phagocytosis also deserves more attention. Therefore, more extensive efforts are necessary to study this stealthy pathogen deeper in order to discover alternative therapies or even more effective vaccines to eliminate brucellosis completely.

## Figures and Tables

**Figure 1 microorganisms-10-02003-f001:**
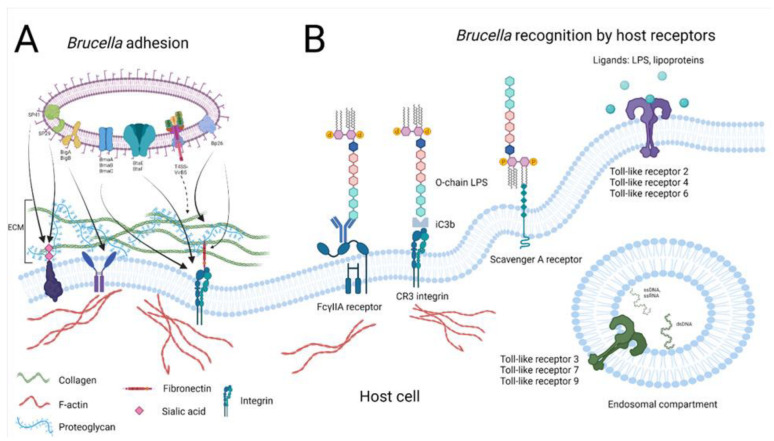
*Brucella* adhesion and recognition. (**A**) *Brucella* expresses various adhesins such as SP29, SP41, BigA, BigB, BmaA, BmaB, BmaC, BtaE, BtaF, Bp26, and T4SS-VirB5. These adhesins enable *Brucella* to adhere to the host cell surface through the extracellular matrix component, including collagen, fibronectin, or sialic acid-binding protein. (**B**) Once tightly adhered to the cell surface, *Brucella* can be detected by binding to various cell receptors. FcγIIA receptor and CR3, opsonic receptors, recognize the O-chain fragment of *Brucella* LPS. Whereas non-opsonic receptors comprise SR-A and TLRs. SR-A recognizes lipid A LPS. TLR2, 4, 6 detect surface molecules of *Brucella* such as LPS and lipoprotein, while TLR3, 7, 9 are responsible for recognition of nucleic acid motifs.

**Figure 2 microorganisms-10-02003-f002:**
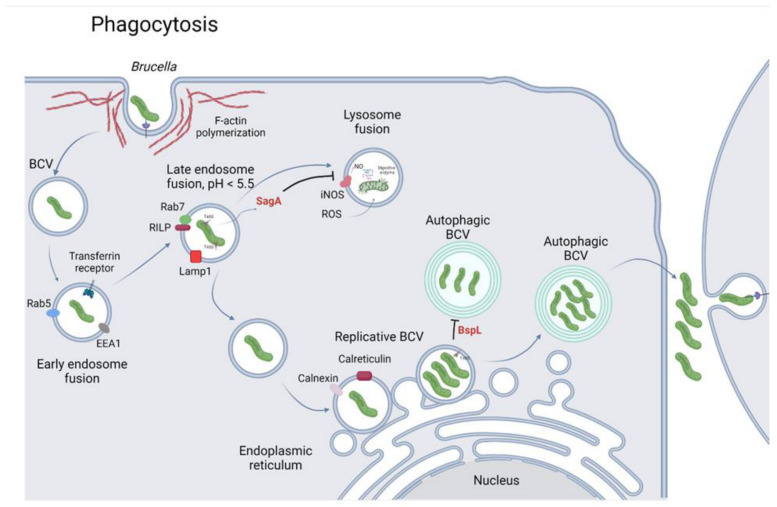
Intracellular trafficking cycle of *Brucella* in host cells. After recognizing *Brucella* by different receptors, host cells activate the signaling pathway resulting in F-actin polymerization at the binding site with bacteria. Then the BCVs undergo fusion with early, late endosome, and finally lysosome. At the last phase of BCVs’ maturation, *Brucella* is killed by nitric oxide, ROS, and digestive enzymes that localize in the phagolysosome compartment. However, *Brucella* has the ability to avoid the phagolysosome fusion to reach its intracellular niche at ER. *Brucella* SagA protein is secreted to interfere with the interaction between BCVs and lysosomes. Once localizing in ER, *Brucella* can proliferate and mature into aBCVs. This aBCVs formation depends on distinct autophagy proteins including ULK1 and Beclin 1, which contribute to bacterial egress and the formation of infection foci resulting in dissemination to other cells and tissues. Besides, *Brucella* activates a T4SS-BspL effector to delay the formation of aBCVs, and ultimately reach maximum proliferation.

**Table 1 microorganisms-10-02003-t001:** *Brucella* functional proteins in interaction with host cells.

*Brucella* Proteins	Functions
Cyclic β-1,2-glucan	Important for circumventing host cell defenses, and modulate lipid raft organization
VirB T4SS	Mediating intracellular survival and circumventing host immune responses
SP29, SP41	Sialic acid-binding proteins, in bacterial adherence
BigA, BigB	Proteins containing the immunoglobulin-like domain, in bacterial adherence
BmaA, BmaB, BmaC	The monomeric autotransporters, in bacterial adherence
BtaE, BtaF	The trimeric autotransporters, in bacterial adherence
Bp26	Collagen, vitronectin-binding protein, in bacterial adherence
VirB5	Effector of a well-known *Brucella* virulence factor T4SS, in bacterial adherence
SagA	A lysozyme-like protein SagA identified as a muramidase
VceC, VecA	T4SS, related to host autophagy and apoptosis
BtpA, BtpB	Modulate host immunity and energy metabolism
BvrS/R	Transcriptionally regulates T4SS VirB
VPS35, VPS26A	Related to the evasion of the lysosomal degradative pathway
BspL	Delay the formation of aBCVs that benefit the optimal intracellular replication before disseminating to other cells
BspJ	A nucleomodulin, directly or indirectly regulates host cell apoptosis to complete its intracellular cycle
DnaK, ClpB	Heat shock proteins, role in bacterial resistance against stresses

## Data Availability

Not applicable.

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
