# Peer review of "Brucella Phagocytosis Mediated by Pathogen-Host Interactions and Their Intracellular Survival"

_microorganisms, 2022, doi:10.3390/microorganisms10102003_

Round 1

Reviewer 1 Report

This is an extensive review of the field, well written and comprehensive.  Not very novel, but maybe such a review is good because this field is not well covered in the scientific literature

Author Response

Sep, 2022

Dear Dr. Byron Liu,

Editor of Microorganisms Journal

Enclosed hereby, please find our revised manuscript (ID: microorganisms-1891262) entitled: “Brucella phagocytosis mediated by pathogen-host interactions and their intracellular survival”.

We would like to thank the Editors and the reviewers for their constructive comments which we took into consideration when revising our manuscript. All of the comments raised by the reviewers have been addressed in detail in our efforts to improve our manuscript and all the changes that we made in response to the reviewers’ comments are highlighted in the yellow text in the revised manuscript. A point-by-point response to the reviewers’ comments follows on the accompanying pages.

We hope that the revised manuscript is now acceptable for publication in Microorganisms Journal. Please contact me with any questions concerning the manuscript. I can be reached at +82-55-772-2359 or by e-mail at kimsuk@gnu.ac.kr.

With regards,

Suk Kim, Ph. D.

College of Veterinary Medicine,

Gyeongsang National University,

Jinju, 660-701, Republic of Korea

Author's Responses to the Reviewer's Comments

ID: microorganisms-1891262

Title: Brucella phagocytosis mediated by pathogen-host interactions and their intracellular survival

Authors: Tran X.N. Huy, Trang T. Nguyen, Heejin Kim, Alisha W.B. Reyes, Suk Kim.

Comments from the reviewers

# Reviewer 1

This is an extensive review of the field, well written and comprehensive.  Not very novel, but maybe such a review is good because this field is not well covered in the scientific literature

Authors answer to Review 1: We are thankful for the reviewer’s comment. In fact, our study is not very novel. We reviewed and updated new research on Brucella phagocytosis and the interaction between Brucella and host cells in the context of intracellular killing and growth. Compared to other bacteria such as Mycobacterium, Salmonella, or Helicobacterium, there are still many gaps in research related to Brucella pathogenesis and treatment. Therefore, these kinds of reviews should be conducted to provide comprehensive insight about Brucella infection.

# Reviewer 2

Q1: The review of Tran Huy et al. focuses on the important stage of the Brucella during the pathogenesis. The first part of the review is depicting bacterial adhesion, and which proteins are crucial for the invasion process. Here, the proteins are listed, however more clear picture of the events is missing (maybe in a form of a figure). Which receptors from the host site are targeted?  Authors are very generally giving the information about Toll-like receptors and complement factors, however, such I find as general for most of the intracellular Gram-negative bacteria.  Are the listed bacterial proteins host-specific? Which are specific ligand-receptors? The figures are missing, I cannot judge their quality/content.

Authors answer to Q1: There are various kinds of receptors related to Brucella phagocytosis. Among them, scavenger receptor A and Toll-like receptors are well described as prominent receptors related to Brucella phagocytosis in professional and non-professional phagocytes. While the research involved in the opsonized receptors until now has been limited in the context of Brucella phagocytosis. Scavenger receptor A is well known as a receptor for lipid A fraction of Brucella LPS. Interestingly, TLRs are well characterized in recognizing Brucella. In particular, TLR3, 7, and 9 recognize Brucella ­nucleic acid structure motifs, while TLR2, 4, and 6 can detect different membrane components of Brucella. In Figure 1, we showed the Brucella adhesion and recognition by host receptors but did not specify exactly host cells. This is due to different cells can express the same receptors for specific Brucella ligands. For example, the study by Watanabe et al. showed TLR2 could recognize Brucella lipoprotein in trophoblast giant cells. Similarly, the interaction between dendritic cells TLR2 with Brucella lipoprotein Omp19 is related to Brucella phagocytosis. Moreover, LPS is a potential ligand for phagocytosis receptors, in particular, the lipid A of LPS is conserved among a wide range of gram-negative bacteria and generally recognized by TLR4 of host cells.

References:

Watanabe, K.; Shin, E.K.; Hashino, M.; Tachibana, M.; Watarai, M. Toll-like receptor 2 and class B scavenger receptor type I are required for bacterial uptake by trophoblast giant cells. Mol Immunol 2010, 47, 1989-1996.

Avila-Calderón, E.D.; Flores-Romo, L.; Sharon, W.; Donis-Maturano, L.; Becerril-García, M.A.; Arreola, M.G.A. Dendritic cells and Brucella spp. interaction: the sentinel host and the stealthy pathogen. Folia Microbiologica 2020, 65, 1-16.

Matsuura, M. Structural modifications of bacterial lipopolysaccharide that facilitate Gram-negative bacteria evasion of host innate immunity. Front Immun 2013, 4: 109.

Q2: The conclusion has to be less repetitive with general information (the first paragraph I find useless).

Authors answer to Q2: We agree with the reviewer’s comment. However, we tried to summarize only the main idea throughout this study. In this study, we made two figures. One figure each describes information from 2 - 3 sections in the body text. Therefore we need this first paragraph of the conclusion section to cite these figures.

Q3: Moreover, I have several minor comments:

- the article needs extensive language editing. There are some general wording issues where I (possibly as a matter of preference) would advise the use of different wording.

Authors answer to Q3: As per reviewer’s comment, we changed the sentences “The transmission of infection to humans is primarily via direct contact with the animals, handling of contaminated tissues, in turn suggesting that agricultural professions and consumption of unpasteurized milk products are the established risk factors for human brucellosis [1]” by “The transmission of infection to humans is primarily via direct contact with the animals, handling of contaminated tissues, and consumption of unpasteurized milk products [1]” in the Introduction section (page 3, lines 47 – 49).

We also changed the sentence “Four steps are needed for Brucella to infect the host: adherence, internalization, intracellular growth, and dissemination within the host [7].” by “Four steps are essential for Brucella to infect the host: adherence, internalization, intracellular growth, and dissemination within the host [7].” in the Introduction section (page 4, lines 87 – 88).

Q4: Introduction is too long and contains information which suits more for the main part. Also, assume the introduction to be relatively incoherent.

Authors answer to Q4: As per the reviewer’s comment, we deleted sentences “The pathogen's entry mechanisms involve lipid raft, adhesin, and opsonin-dependent processes. Brucella enters the host through the mucosal membranes of the respiratory and digestive tracts, internalized by local professional phagocytes, and then moved to the closest draining lymph nodes that lead to subsequent dissemination to different organs of the reticuloendothelial system, which include the lung, spleen, liver and bone marrow while the pathogens display a strong tropism for placental trophoblasts and for mammary glands in pregnant animals [1]” in the Introduction section (page 4, line 88).

In addition, we changed the sentence “This review mainly focuses on the phagocytosis of Brucella spp. into the host cells, which currently there is still minimal information available to completely describe this Brucella’s interaction with its target cells and tissues.” by “ This review mainly focuses on the phagocytosis of Brucella spp. into the host cells as well as its intracellular growth in the macrophage cell model, which currently there is still minimal information available to completely describe this Brucella’s interaction with its target cells and tissues.” in the Introduction section (pages 4 – 5, lines 88 – 91).

Q5: Part 2: The general phagocytosis information I am recommending to remove - usually, readers who are searching scientific paper about Brucella have such knowledge.

Authors answer to Q5: As per reviewer’s reccomendation, we deleted the paragraph “Phagocytosis is an essential cellular mechanism for internalizing particles including microorganisms, foreign substances, and apoptotic cells with sizes larger than 0.5 µm. Phagocytosis can be classified into two types depending on the cell types and their functions in cellular physiological processes. The first is the specialized group of cells that can be considered as professional phagocytes including macrophages, neutrophils, monocytes, and DCs [12]. The phagocytosis of these immune cells plays a role as a sentinel of host innate and adaptive defense. Whereas fibroblasts, epithelial cells, and endothelial cells can not eliminate pathogens but play a critical role in eliminating dead cells and maintaining cell homeostasis, these are called non-professional phagocytes. In the context of Brucella spp. infection, the phagocytosis process consists of three phases: i) pathogen recognition, ii) engulfment of the pathogen resulting in Brucella­-containing vacuoles (BCVs) formation, and iii) host intracellular killing or bacterial intracellular growth [13].” In the “2. Brucella phagocytosis and intracellular survival” section (page 5, line 93).

Q6. A table with the important protein involved in the adhesion, pathogenesis etc would be a nice  benefit of a paper

Authors answer to Q6: As per reviewer’s suggestion, we added a sentence “Furthermore, all Brucella functional proteins related to the interaction between Brucella and host cells is summarized in table 1.” in the Conclusion section (page 15, lines 314 – 316).

And we added more the Table 1 as follows in the Table and Figures legends section (page 29 – 30, lines 647 – 648)

Brucella proteins

Functions

Cyclic β-1,2-glucan

Important for circumventing host cell defenses, and modulate lipid raft organization.

VirB T4SS

Mediating intracellular survival and circumventing host immune responses

SP29, SP41

Sialic acid-binding proteins, in bacterial adherence

BigA, BigB

Proteins containing the immunoglobulin-like domain, in bacterial adherence

BmaA, BmaB, BmaC

the monomeric autotransporters, in bacterial adherence

BtaE, BtaF

the trimeric autotransporters, in bacterial adherence

Bp26

collagen, vitronectin-binding protein, in bacterial adherence

VirB5

effector of a well-known Brucella virulence factor T4SS, in bacterial adherence

SagA

A lysozyme-like protein SagA identified as a muramidase

VceC, VecA

T4SS, related to host autophagy and apoptosis

BtpA, BtpB

Modulate host immunity and energy metabolism

BvrS/R

Transcriptionally regulates T4SS VirB

VPS35, VPS26A

Related to the evasion of the lysosomal degradative pathway

BspL

Delay the formation of aBCVs that benefit the optimal intracellular replication before disseminating to other cells

BspJ

A nucleomodulin, directly or indirectly regulates host cell apoptosis to complete its intracellular cycle

DnaK, ClpB

Heat shock proteins, role in bacterial resistance against stresses

Q7: Also the list of abbreviations used in the paper will help readers easier orientation in the review

Authors answer to Q7: As per reviewer’s suggestion, we added a list of abbreviation in the List of abbreviation section (pages 31 – 32, lines 670 – 688).

# Reviewer 3

Brucellosis is a very serious zoonotic infection, which poses a threat to global public health security. This review has certain guiding significance for revealing the pathogenic mechanism of brucellosis. However, the whole article should understand one thing: Brucella belongs to facultative intracellular parasitic bacteria, and the intracellular bacteria in the manuscript must be changed.

Q1Brucella belongs to facultative intracellular parasitic bacteria. This concept, including the abstract and text, should be changed.

Authors answer to Q1: As per the reviewer’s comment. We changed the sentence “Brucella is an intracellular bacterium” by “Brucella is a facultative intracellular bacterium” in the Abstract section (page 2, lines 26 – 27). In addition, we also changed the sentence “Brucella sp. is an intracellular pathogen” by “Brucella sp. is a facultative intracellular pathogen” in the Conclusion section (page 14, line 305). 

Q2: Macrophages and placental trophoblast cells are the main target cells of Brucella infection, and they can also survive in dendritic cells. Therefore, the author should clarify the intracellular parasitic mechanism of Brucella in these three types of cells by classification

Authors answer to Q2: As per the reviewer’s comment. We added new sentences “Dendritic cells are well-known antigen-presenting cells, also considered safe heaven for Brucella growth. Brucella can interfere their maturation leading to inhibit the antigen processing and presentation, that circumvent the host immune responses. Moreover, Brucella can infect the animal placenta resulting in abortion. In particular, it can replicate in the placental trophoblasts, where produce erythritol. Indeed, the erythritol utilization is one of Brucella virulence factors [11].” in the Introduction section (page 4, lines 81 – 87).

We also added sentences “In addition, TLR2 also plays a critical role in Brucella invasion into the trophoblast giant cells [46]. These suggested that Brucella can internalize into different types of cells in the same way” in the “Host receptors” section (page 8, lines 169 – 171).

And we also added a new reference at page 203, lines 509 – 512:

  1. Watanabe, K.; Shin, E.K.; Hashino, M.; Tachibana, M.; Watarai, M. Toll-like receptor 2 and class B scavenger receptor type I are required for bacterial uptake by trophoblast giant cells. Mol Immunol 2010, 47, 1989-1996.

Q3: In particular, figure 1 should be shown around this idea, and the reader will understand it at a glance.

Authors answer to Q3: Brucella can infect different types of cells in the same ways, including Toll-like receptors and scavenger receptors. Indeed, a study by Watanabe et al. showed that the trophoblast giant cells could display phagocytic activity through TLR2 against Brucella infection. Furthermore, TLRs are also related to the phagocytosis of Brucella in dendritic cells. Therefore, we described the Brucella adhesion and recognition in the host cells using Figure 1 as representative of the target cells of Brucella infection.

References:

Watanabe, K.; Shin, E.K.; Hashino, M.; Tachibana, M.; Watarai, M. Toll-like receptor 2 and class B scavenger receptor type I are required for bacterial uptake by trophoblast giant cells. Mol Immunol 2010, 47, 1989-1996.

Avila-Calderón, E.D.; Flores-Romo, L.; Sharon, W.; Donis-Maturano, L.; Becerril-García, M.A.; Arreola, M.G.A. Dendritic cells and Brucella spp. interaction: the sentinel host and the stealthy pathogen. Folia Microbiologica 2020, 65, 1-16.

Q4Brucella can use autophagy to regulate the process of its intracellular survival, which is a very frontier hotspot and should be summarized, at least reflected.

Authors answer to Q4: As per the reviewer’s comment, we added sentences “Autophagy-initiation proteins ULK1, Beclin 1, ATG14L and PI3-kinase activity is required for the aBCVs formation. This event is exploited by Brucella ­­for cell-to-cell spreading [84,85]” in the “2.3.2. Brucella intracellular survival” section (page 14, lines 292 – 294).

Furthermore, in agreement with the reviewer’s comment. We also recognize that the clarification of interaction between Brucella and endoplasmic reticulum as well as autophagy process will provide comprehensive insight. Therefore, we will write another review paper about the interaction between Brucella and ER as well as autophagy.

And we added two new references at pages 28 – 29, lines 629 – 633:

  1. Brumell, J.H. Brucella ‘‘Hitches a Ride’’ with Autophagy. Cell Host Microbe 2012, 11, 2-4.
  2. Starr, T.; Child, R.; Wehrly, T.D.; Hansen, B.; Hwang, S.; Lo´ pez-Otin, C. et al. Selective subversion of autophagy complexes facilitates completion of the Brucella intracellular cycle. Cell Host Microbe 2012, 11, 33-45.

Q5: The language of the article still needs to be improved, and some sentences are difficult to understand.

Authors answer to Q5: As per reviewer’s comment, we changed the sentences “The transmission of infection to humans is primarily via direct contact with the animals, handling of contaminated tissues, in turn suggesting that agricultural professions and consumption of unpasteurized milk products are the established risk factors for human brucellosis [1]” by “The transmission of infection to humans is primarily via direct contact with the animals, handling of contaminated tissues, and consumption of unpasteurized milk products [1]” in the Introduction section (page 3, lines 47 – 49).

We also changed the sentence “Four steps are needed for Brucella to infect the host: adherence, internalization, intracellular growth, and dissemination within the host [7].” by “Four steps are essential for Brucella to infect the host: adherence, internalization, intracellular growth, and dissemination within the host [7].” in the Introduction section (page 4, lines 87 – 88).

Reviewer 2 Report

The review of Tran Huy et al. focuses on the important stage of the Brucella during the pathogenesis. The first part of the review is depicting bacterial adhesion, and which proteins are crucial for the invasion process. Here, the proteins are listed, however more clear picture of the events is missing (maybe in a form of a figure). Which receptors from the host site are targeted?  Authors are very generally giving the information about Toll-like receptors and complement factors, however, such I find as general for most of the intracellular Gram-negative bacteria.  Are the listed bacterial proteins host-specific? Which are specific ligand-receptors? The figures are missing, I cannot judge their quality/content.

The conclusion has to be less repetitive with general information (the first paragraph I find useless).

Moreover, I have several minor comments:

- the article needs extensive language editing. There are some general wording issues where I (possibly as a matter of preference) would advise the use of different wording.

Introduction is too long and contains information which suits more for the main part. Also, assume the introduction to be relatively incoherent.

Part 2:

- the general phagocytosis information I am recommending to remove - usually, readers who are searching scientific paper about Brucella have such knowledge.

-a table with the important protein involved in the adhesion, pathogenesis etc would be a nice  benefit of a paper

-also the list of abbreviations used in the paper will help readers easier orientation in the review

Author Response

(The authors gave the same response as above.)

Reviewer 3 Report

Brucellosis is a very serious zoonotic infection, which poses a threat to global public health security. This review has certain guiding significance for revealing the pathogenic mechanism of brucellosis. However, the whole article should understand one thing: Brucella belongs to facultative intracellular parasitic bacteria, and the intracellular bacteria in the manuscript must be changed.

1) Brucella belongs to facultative intracellular parasitic bacteria. This concept, including the abstract and text, should be changed.

2) Macrophages and placental trophoblast cells are the main target cells of Brucella infection, and they can also survive in dendritic cells. Therefore, the author should clarify the intracellular parasitic mechanism of Brucella in these three types of cells by classification. In particular, figure 1 should be shown around this idea, and the reader will understand it at a glance.

3) Brucella can use autophagy to regulate the process of its intracellular survival, which is a very frontier hotspot and should be summarized, at least reflected.

4) The language of the article still needs to be improved, and some sentences are difficult to understand.

Author Response

(The authors gave the same response as above.)

Round 2

Reviewer 2 Report

The paper was significantly improved, however, the name of the pathogen in the table is not in italics, plus I cannot find figures - they are not attached to submitted document? That's why, I cannot judge their quality and relevance.

Author Response

Sep, 2022

Dear Dr. Byron Liu,

Editor of Microorganisms Journal

Enclosed hereby, please find our revised manuscript (ID: microorganisms-1891262) entitled: “Brucella phagocytosis mediated by pathogen-host interactions and their intracellular survival”.

We would like to thank the Editors and the reviewers for their constructive comments which we took into consideration when revising our manuscript. All of the comments raised by the reviewers have been addressed in detail in our efforts to improve our manuscript and all the changes that we made in response to the reviewers’ comments are highlighted in the yellow text in the revised manuscript. A point-by-point response to the reviewers’ comments follows on the accompanying pages.

We hope that the revised manuscript is now acceptable for publication in Microorganisms Journal. Please contact me with any questions concerning the manuscript. I can be reached at +82-55-772-2359 or by e-mail at kimsuk@gnu.ac.kr.

With regards,

Suk Kim, Ph. D.

College of Veterinary Medicine,

Gyeongsang National University,

Jinju, 660-701, Republic of Korea

Author's Responses to the Reviewer's Comments

ID: microorganisms-1891262

Title: Brucella phagocytosis mediated by pathogen-host interactions and their intracellular survival

Authors: Tran X.N. Huy, Trang T. Nguyen, Heejin Kim, Alisha W.B. Reyes, Suk Kim.

Comments from the reviewers

# Reviewer 2

The paper was significantly improved, however, the name of the pathogen in the table is not in italics, plus I cannot find figures - they are not attached to submitted document? That's why, I cannot judge their quality and relevance.

Q1: the name of the pathogen in the table is not in italics

Authors answer to Q1: As per the reviewer’s comment, we checked and revised Table 1 in the Table and Figures legends (page 29 – 30, lines 648 – 649)

Table 1. Brucella functional proteins in interaction with host cells

Brucella proteins

Functions

Cyclic β-1,2-glucan

Important for circumventing host cell defenses, and modulate lipid raft organization.

VirB T4SS

Mediating intracellular survival and circumventing host immune responses

SP29, SP41

Sialic acid-binding proteins, in bacterial adherence

BigA, BigB

Proteins containing the immunoglobulin-like domain, in bacterial adherence

BmaA, BmaB, BmaC

The monomeric autotransporters, in bacterial adherence

BtaE, BtaF

The trimeric autotransporters, in bacterial adherence

Bp26

Collagen, vitronectin-binding protein, in bacterial adherence

VirB5

Effector of a well-known Brucella virulence factor T4SS, in bacterial adherence

SagA

A lysozyme-like protein SagA identified as a muramidase

VceC, VecA

T4SS, related to host autophagy and apoptosis

BtpA, BtpB

Modulate host immunity and energy metabolism

BvrS/R

Transcriptionally regulates T4SS VirB

VPS35, VPS26A

Related to the evasion of the lysosomal degradative pathway

BspL

Delay the formation of aBCVs that benefit the optimal intracellular replication before disseminating to other cells

BspJ

A nucleomodulin, directly or indirectly regulates host cell apoptosis to complete its intracellular cycle

DnaK, ClpB

Heat shock proteins, role in bacterial resistance against stresses

Q2: I cannot find figures - they are not attached to submitted document? That's why, I cannot judge their quality and relevance.

Authors answer to Q2: As per the reviewer’s comment, we already attached all figures when submitting the revised manuscript. In addition, we also insert all figures here for easy reading as below:

Figure 1. Brucella adhesion and recognition. (A) Brucella expresses various adhesins such as SP29, SP41, BigA, BigB, BmaA, BmaB, BmaC, BtaE, BtaF, Bp26, and T4SS-VirB5. These adhesins enable Brucella to adhere to the host cell surface through the extracellular matrix component, including collagen, fibronectin, or sialic acid-binding protein. (B) Once tightly adhered to the cell surface, Brucella can be detected by binding to various cell receptors. FcγIIA receptor and CR3, opsonic receptors, recognize the O-chain fragment of Brucella LPS. Whereas non-opsonic receptors comprise SR-A and TLRs. SR-A recognizes lipid A LPS. TLR2, 4, 6 detect surface molecules of Brucella such as LPS and lipoprotein, while TLR3, 7, 9 are responsible for recognition of nucleic acid motifs.

Figure 2. Intracellular trafficking cycle of Brucella in host cells. After recognizing Brucella by different receptors, host cells activate the signaling pathway resulting in F-actin polymerization at the binding site with bacteria. Then the BCVs undergo fusion with early, late endosome, and finally lysosome. At the last phase of BCVs maturation, Brucella is killed by nitric oxide, ROS, and digestive enzymes that localize in the phagolysosome compartment. However, Brucella has the ability to avoid the phagolysosome fusion to reach its intracellular niche at ER. Brucella SagA protein is secreted to interfere with the interaction between BCVs and lysosomes. Once localizing in ER, Brucella can proliferate and transfer into aBCVs which release bacteria from infected cells and disseminate to other cells and tissues. Besides, Brucella activates a T4SS-BspL effector to delay the formation of aBCVs, and ultimately reach maximum proliferation.

# Reviewer 3

The author has made a relatively detailed revision according to the previous review comments, and the quality of the manuscript has been significantly improved. The section of Brucella mediated autophagy requires another reference: Hanwei Jiao et al, 2020. There are some small mistakes in the article, such as CDc 42? CDC42?

Q1: The section of Brucella mediated autophagy requires another reference: Hanwei Jiao et al, 2020

Authors answer to Q1: As per the reviewer’s comment. We added a new reference for the section related to Brucella-mediated autophagy in the Reference section (page 29, lines 632 – 634).

Reference:

  1. Hanwei, J.; Nie, X.; Zhu, H.; Li, B.; Pang, F.; Yang, X.; Cao, R.; Yang, X.; Zhu, S.; Peng, D. miR-146b-5p Plays a Critical Role in the Regulation of Autophagy in∆ per Brucella melitensis-Infected RAW264. 7 Cells. BioMed research international 2020, 2020.

And we also changed the sentence “Autophagy-initiation proteins ULK1, Beclin 1, ATG14L and PI3-kinase activity is required for the aBCVs formation. This event is exploited by Brucella ­­for cell-to-cell spreading [84,85].” by “Autophagy-initiation proteins ULK1, Beclin 1, ATG14L and PI3-kinase activity is required for the aBCVs formation. This event is exploited by Brucella ­­for cell-to-cell spreading [84-86].” In the Discussion section (page 14, lines 291 – 293).

Q2: There are some small mistakes in the article, such as CDc 42? CDC42?

Authors answer to Q2: As per the reviewer’s comment. We changed into “Cdc42” for all in the manuscript, in particular, in the “2.2.1. Bacterial engulfment” section (page 9, lines 191 and 196).

Reviewer 3 Report

The author has made a relatively detailed revision according to the previous review comments, and the quality of the manuscript has been significantly improved. The section of Brucella mediated autophagy requires another reference: Hanwei Jiao et al, 2020. There are some small mistakes in the article, such as CDc 42? CDC42?

Author Response

(The authors gave the same response as above.)
